# The Promises, Challenges, and Opportunities of Omics for Studying the Plant Holobiont

**DOI:** 10.3390/microorganisms10102013

**Published:** 2022-10-12

**Authors:** Dana L. Carper, Manasa R. Appidi, Sameer Mudbhari, Him K. Shrestha, Robert L. Hettich, Paul E. Abraham

**Affiliations:** 1Biosciences Division, Oak Ridge National Laboratory, Oak Ridge, TN 37831, USA; 2Graduate School of Genome Science and Technology, University of Tennessee-Knoxville, Knoxville, TN 37996, USA

**Keywords:** genomics, transcriptomics, proteomics, metabolomics, plant–microbe interaction, plant holobiont

## Abstract

Microorganisms are critical drivers of biological processes that contribute significantly to plant sustainability and productivity. In recent years, emerging research on plant holobiont theory and microbial invasion ecology has radically transformed how we study plant–microbe interactions. Over the last few years, we have witnessed an accelerating pace of advancements and breadth of questions answered using omic technologies. Herein, we discuss how current state-of-the-art genomics, transcriptomics, proteomics, and metabolomics techniques reliably transcend the task of studying plant–microbe interactions while acknowledging existing limitations impeding our understanding of plant holobionts.

## 1. Introduction

Technological advancements in high-throughput genome sequencing and mass spectrometry have transformed biological sciences. Today, there is an ever-growing list of revolutionary approaches with the suffix-omics [1] that extends beyond those derived from the central dogma, i.e., genomics, transcriptomics, proteomics, and metabolomics. The application of these technologies in plant sciences has transformed our understanding of plant–microbe interactions [2]. Since the publication of the *Arabidopsis thaliana* microbiome [3], providing our first detailed look at this complex microbial world, scientists around the world have revealed how plants and microbes have their own sophisticated communication networks and division of labor that is subject to selection in alternative environments [4].

In recent years, it is becoming increasingly apparent that plant phenotypes are a result of the combined expression of the host and associated microbial genomes, leading to the popularization of the ‘holobiont’ theory [5]. As a result, the concept of a plant holobiont, an assemblage of highly cooperative and minimally conflicting plant–microbe interactions, is becoming more frequently used in plant sciences [6]. Today, the field of plant holobiont research invokes the study of interlinkages between plant and individual microbial behavior and evolution to understand how functionally integrated they are or how natural selection operates on them [7]. While the concept of holobionts can seem unnecessarily complex at times, ecological functions provided by microbes are now regarded as an important feature of plant fitness [8]. Thus, unraveling the complexity of holobionts promises to deliver innovations in plant ecosystem productivity for sustainable agriculture [9,10] by rooting out stochasticity and fortifying predictability [11].

The paradigm shift towards the increasingly recognized concept of plant holobionts introduces new questions to be answered [12,13] and, consequently, a new theoretical framework for omic technologies to follow. Conceptually, due to inherently dynamic biotic-abiotic interactions with the environment, the plant holobiont structure and composition will experience adaptive cycles of expansion, consolidation, and resilience [11]. Under such premises, plants and microbes “work together” to continually adapt the plant holobiont to perpetually changing environmental conditions. Efforts to assess the strength of genetic, molecular, and metabolic relationships between plants and microbes across environments would contribute to a more accurate view of the plant holobiont at evolutionary and ecological scales. Therefore, in this perspective, we discuss the promises and challenges of omics technologies in studying plant holobionts and how omics information can be aggregated across these adaptive cycles to effectively understand the long-term fidelity of plant–microbe interactions and discern their connectivity to ecological functions in plant holobionts (Figure 1).

## 2. Advancements in Omics Are Key to Defining Plant Holobionts

Major scientific breakthroughs in the study of plant–microbe interactions are driven by technological advancements that facilitate cost-efficient, high-throughput analysis of DNA, RNA, proteins, and metabolites. In the past decade, high-throughput sequencing technologies, e.g., Illumina (https://www.illumina.com/, accessed on 15 September 2022) Pacific Biosciences (https://www.pacb.com/, accessed on 15 September 2022) and Oxford Nanopore Technologies (https://nanoporetech.com/, accessed on 15 September 2022), have fostered rapid progress in the field of plant–microbe research by delivering insights into relevant genetic and genomic expression signatures. Innovations in tandem mass spectrometry [14] have provided access to how those genomic signatures are translated to proteins and their subsequent metabolic products. Today, as a result of these advances, we are now able to answer questions at astonishing levels of mechanistic detail.

Moving forward, our understanding of the plant holobiont will require host-centered omic strategies paired with commonly used microbial-focused techniques, such as amplicon sequencing and meta-omics [15]. Over the past decade, microbiome sequencing and analysis has improved our understanding of the structure and diversity of the microbial world that grows in (endosphere) and on above-(phyllosphere) and below-ground (rhizosphere) plant tissues. Early efforts that used standardized protocols for 16S ribosomal RNA (rRNA) sequencing expanded our worldly understanding of microbial diversity [16,17], its extent and limit, and how a plant host and/or environment selects for specific taxonomic and phylogenetic composition [18]. Similar to 16S rRNA sequencing, DNA barcoding of Internal Transcribed Spacer (ITS) region of the nuclear DNA has been a key molecular method for our understanding of fungal diversity [19,20,21,22].

Today, next-generation sequencing technologies are increasingly used in attempts to identify key or “core” microbiome members that consistently engage with plants directly; a key ecological parameter in holobiont theory. The most actively applied approach to define a core microbiome prioritizes membership by taxonomic rank, which is determined by a member’s occupancy and abundance across longitudinal studies [23]. In the simplest way, taxa with relatively high abundance or observed more frequently can be interpreted as core taxa, though conditionally rare species can also play important roles [24]. While high-throughput sequencing using marker genes (e.g., 16S rRNA, ITS or 18S rRNA gene) is being performed at unprecedented spatial and temporal levels [25], these methods lack functional information. Therefore, in addition to taxonomic approaches, it is important to integrate or, at the very least, follow up with functional data (metagenomics [26], metatranscriptomics [27], metaproteomics [28], and metabolomics [29,30,31]) because we are learning that microbiomes having different species can still encode similar functions [32]. It is expected that comparative functional metagenomics combined with other downstream meta-omic methods represents a critical step to our discoveries of interacting mechanisms between plants and microbes that explain consistently defined core microbiome taxa or function [33,34].

As discussed later in this perspective, integrating multi-omics data is inherently difficult because genome expression, transcription, translation, and metabolism all operate on different timescales [35]. As the field of plant sciences trends towards ever-larger data sets with multiple omic layers, state-of-the-art approaches will employ machine-learning and explainable artificial intelligence approaches that serve as a means to classify and interpret key relationships across a multitude of variables (e.g., plant host genotype, microbiome composition and function, environment, time, space, etc.) [36].

## 3. From Genes to Ecosystems: Studying Plant–Microbe Interactions across the Complexity Landscape

Incorporating natural genetic, environmental, and functional complexity into a single experimental framework represents a key challenge for all omic technologies—sparse data combined with methodological limitations can lead to insufficient information or, even worse, misleading biological inferences [37]. Hence, identifying the genetic, molecular, and metabolic factors underpinning emergent plant microbiome-associated phenotypes in the environment is recognized as a daunting task. To address this, plant–microbe interaction studies adopt experimental frameworks that either seek to control or embrace natural complexity (Figure 2) [38,39].

Using a reductionist approach, relatively low levels of genotypic and functional diversity are studied in habitats operating with a highly controlled environment. Under this framework, simplicity and experimental control offers greater interpretability. Today, a major advantage of using a reductionist approach is the ability to develop and use engineered plant–microbe habitats, for example, EcoFab devices [40], to control complexity while seamlessly integrating with omic measurements. On the one hand, the reductionist strategy offers an opportunity to define plant holobionts with exquisite mechanistic detail, providing genetic and/or molecular explanations for plant–microbe interactions. On the other hand, this strategy is not poised for studying higher-order ecological processes and their importance.

Moving to the other side of the complexity-control spectrum, a holistic approach captures ecological interactions with the plant holobiont. Using this experimental framework, large-scale surveys incorporating marker gene or metagenomic sequencing are used to study plant–microbe interactions across complex natural environments. This approach offers an opportunity to disentangle the relative influence of genotypic, environmental, and functional variables and the ecological importance between these variables [41]. Except for a few exemplary examples [42,43], holistic approaches are often without omic information due to cost and labor.

Arguably the next step forward in plant–microbe interaction studies is to integrate reductionist and holistic approaches [44]. Unfortunately, integrating these approaches is not likely achievable for a single research group. Instead, integration takes place across large, interdisciplinary research projects such as several notable efforts supported by the Department of Energy (https://genomicscience.energy.gov/sfas/, accessed on 15 September 2022) and the Earth Microbiome Project (https://earthmicrobiome.org/, accessed on 15 September 2022) [17]. When complete integration is not feasible, it is recommended that careful experimental design be implemented to bridge the knowledge gaps between the reductionist-holistic divide by working inward from both sides of the complexity-control spectrum [38]. In the sections below, we highlight state-of-the-art examples of omics technologies being applied across the complexity-control spectrum and acknowledge key challenges that must be addressed.

### 3.1. Recent Advancements and Current Impediments for Genomics, Transcriptomics, Proteomics, and Metabolomics for Studying the Plant Holobiont

As discussed in the section above, designing an experiment that generates accurate and meaningful results that can be built upon is a challenging task when studying plant holobionts because there are many confounding factors that warrant careful attention. Coordination between research objectives and technology approaches is essential to deploy the appropriate measurements to acquire the desired information. This section aims to highlight the current state-of-the-art research employing omic technologies to better understand plant–microbe interactions as well as notable impediments to their application in complex biological systems.

#### 3.1.1. Genomic and Transcriptomics

In the past decade, considerable efforts have been made using next-generation genomics and transcriptomics to understand how plant genomes influence the presence and function of bacteria and fungi in associated microbiomes [45,46,47,48,49,50,51]. Yet, the extent to which plants exert genetic control over their microbiomes is difficult to disentangle from other natural exogenous stimuli. Natural environments can have biotic and abiotic influences that can mask the effects of host genes that vary between locations and years, demanding multi-year, large-scale field experiments [52,53]. Reductionist approaches using laboratory-controlled conditions can eliminate these confounding factors; however, these studies may also overestimate the influence of certain plant genes and fail to identify genotypic signatures associated with plasticity under context-dependent requirements [54]. Nevertheless, remarkable progress has been made in our understanding of how host genetics drives the composition and function of associated microbiomes [55,56,57].

The contribution of microbial genes to host function and adaptation is similarly dependent on ecological context [58,59]. Yet, the contribution of microbial genetics in holobionts remains difficult to assess simply because the vast majority of microbes lack reference genomes, and this is because most microorganisms are challenging to grow under laboratory conditions. While recent advancements in experimental technologies promise to close this gap using microbial culturomics [60] to isolate and sequence individual genomes, amplicon-based studies (e.g., 16S rRNA, 18S rRNA and ITS) will likely remain the most broadly applied genomic technique to study microbiome diversity and its impact on host function and adaptation [61]. Relevant to the concept of a holobiont, amplicon studies have also begun to shed light on the importance of microbe–microbe interactions within plant communities [61,62]. Yet, as mentioned previously, while amplicon studies are useful for estimating microbial diversity, they fail to provide evidence pertaining to the functional potential and activity of the sampled microbiota. With decreasing costs for massively parallel DNA and RNA sequencing, metagenomic and metatranscriptomic approaches promise to fill this knowledge gap. Today, entire microbial genomes can be reconstructed from metagenome sequencing [63], yielding metagenome-assembled genomes (MAGs) that have increased our functional knowledge of specific microbes within many plant species [64,65,66]. For instance, the study by Xu et al. [66] is an exemplar study demonstrating the utility of genome-resolved metagenomics coupled with downstream reductionist experiments for dissecting plant–microbe interactions in the root-associated microbiome [66]. On the basis of assessing the activity of microbes in mixed communities, isotopic labeling with DNA-based sequencing can be used [67,68,69], but metatranscriptomics remains the most widely adopted technology to assess the functional responses of both plants and microbes to interactions with each other [70,71,72,73] and external environmental stresses [74,75,76].

Moving forward, the combination of host and microbial genomic and transcriptomic information is critically important to improving our understanding of the plant holobiont. In addition to the experimental advancements that further the coverage and resolution of DNA and RNA sequencing in mixed communities [77,78,79,80], innovative computational approaches that improve taxonomic classification [81,82] as well as assembly-based and mapping-based meta-genomic and -transcriptomic profiling are equally important [83,84]. With continual improvements, the integrated study of the genetic features of a plant host alongside that of its associated microbes is becoming a more feasible, though still underdeveloped, approach to understanding plant holobionts [85].

#### 3.1.2. Proteomics

Proteins are considered the central intermediates between a genotype and phenotype, serving as the effectors of function in biology [86]. Currently, it is important to recognize that proteins are no longer considered to be simple translations of genetic code. The extent of chemical diversity proteins can obtain after translation is quite remarkable [87,88], whereby disparate sources of biological variation (e.g., alternative splicing of RNA and post-translational modifications) will affect the fidelity and robustness of a protein structure and function. In general, attempts to compositionally map proteins and their abundances is largely achieved by mass spectrometry-based approaches [89,90]. Over the past several decades, advancements in proteomics have led to new mechanistic insights into how plant hosts recognize their associated microbes and regulate their establishment, persistence, and function [51,91,92]. Beyond the large-scale endeavors that compositionally map and quantify proteome expression changes, the field of proteomics research is recognized as the key data layer to defining the dynamic signal exchange between organisms that allows for recognition between friend and foe [93]. For instance, proteomics has advanced our understanding of early recognition events in the classic example of a plant–microbe interaction—Legume-*Rhizobium* symbiosis [94]. Measuring cellular and subcellular proteomes not only gives information about what happens to a particular host cellular compartment under symbiotic relationships [51,95,96,97,98], but also includes information necessary to monitor signaling events occurring during the early stages of symbiotic interactions [99,100,101,102,103].

With respect to studying the plant holobiont, and similar to the other omics, the ability to integrate host and microbe omics data together is a challenging yet necessary step forward. Unlike animal and human host-microbe systems [104,105,106], only in the past recent years has the field of plant–microbe research started to implement metaproteomics into experimental designs [107,108,109]. This is largely explained by experimental and technical aspects that challenge the depth and coverage of mapping plant-associated metaproteomes, especially those that are endophytic. As such, early attempts to apply metaproteomics so far largely consist of reductionist approaches to study plant–microbe [110] and microbe–microbe interactions [111]. Moving forward, innovations to improve the experimental [112,113,114,115], technical [116], and computational extraction and analysis of metaproteome data [28,117] from more complex environmental matrices, such as native soil, will be crucial to advancing our understanding of plant holobionts.

#### 3.1.3. Metabolomics

Within plant holobionts, metabolites are the immediate effectors underlying the basic processes of recognition and communication between organisms and they are a currency and commodity shared between symbiotic, commensal, parasitic and pathogenic relationships. Based on genome predictions, we know that plants, and even their associated microbes, have the ability to produce thousands of molecules that together interact with and influence ecosystems [118]. Currently, mass spectrometry-based metabolomics is one of the key technologies used to characterize these diverse complex chemical inventories [119]. In the past decade, the improved accessibility and knowledge obtained from metabolomics has been crucial to understanding how plant metabolites shape their microbial communities and how microbially derived molecules affect plant hosts and ecosystems [120,121,122]. In general, metabolomics is frequently applied to (i) characterize plant root exudation and its impact on microbes in the surrounding environment and (ii) the effect of associated microbes on host metabolism.

Direct analysis of plant–microbe relationships in situ could provide the most relevant data for understanding these biological phenomena, but this can be incredibly difficult to reproduce, replicate and standardize because of high variability in environmental factors, such as soil properties, and it can be easily confounded by the surrounding breakdown of unrelated organisms and other biomaterials [123]. Therefore, the vast majority of metabolomic research employs a reductionist approach. To date, the rhizospheric effect on microbiome composition and function has been studied mostly in sterile or (semi)sterile soil or artificial environmental matrices and habitats, such as hydroponic growth systems [124,125,126]. Because of major technical advancements in mass spectrometry, we now know a great deal about root exudate composition in model species such as *Arabidopsis* [127,128,129], maize [130], and rice [131,132] and their effects on associated microbiota [133,134,135]. Today, there are still a small number of examples of metabolomics being applied to study root exudates in either field [136,137] or greenhouse soils [138,139]; however, recent experimental advancements promise to address challenges related to non-sterile soil matrices [140].

Crucial to understanding metabolic linkages in plant holobionts, metabolomics can provide a deep appreciation and understanding of how microbially derived metabolites impact plant phenotypes [141,142,143], either by specific pairwise plant–microbe interactions [144,145,146,147] or microbiome-driven changes in plant metabolomes [148]. At present, there is still little, or no effort made to differentiate the origin of metabolites analyzed in plant–microbe co-cultures and this makes linking metabolome information to a particular phenotype challenging. Attempts to distinguish metabolites will benefit our understanding of plant–microbe and microbe–microbe metabolic interactions and thus the use of stable-isotope labeling [149,150] is considered as a promising technical advancement towards our understanding of plant holobionts.

#### 3.1.4. Integrative Systems Biology

Combining multiple-omic technologies can be challenging due to the extent of data, lack of consensus between data types, and the different scales at which each technology measures the plant holobiont. When successfully integrated, multi-omic studies offer unprecedented insights into the mechanistic interplay between plants and microbes [137,151,152,153]. Advancements in computational tools and deep-learning applications that account for the increased and varied data types are improving interpretability (reviewed in [154,155,156] while network analyses continue to be a useful approach to analyze the integration of multiple data set types [157,158]. Moving forward, efforts by the research community to extend the utility and accessibility of computations tools and bioinformatic workflows will be a key factor in our scientific advancement of the plant holobiont concept into practical applications. For instance, the open-source data science platform KBase (http://kbase.us, accessed on 15 September 2022) [159], a freely available community resource offering a suite of tools and workflows designed as a “one-stop-shop” to integrate and analyze complex data types, has seen tremendous growth in utilization by the research community. Equally important to the accessibility of tools is the underlying data. Effective data sharing, using FAIR principles [160], is important for this growing research community. While proper data sharing for DNA and RNA sequences is becoming more routine, the ability to “FAIRify” mass spectrometry data from proteomics and metabolomics still represents a significant challenge for the research community. The development of data infrastructures such as GNPS (https://gnps.ucsd.edu/, accessed on 15 September 2022) [161] represented, and continues to be, an exciting and important step in the right direction for capturing and retaining knowledge obtained by mass spectrometry.

## 4. Conclusions

Understanding the mechanistic principles central to plant holobiont theory provides an opportunity to predict and augment beneficial and detrimental plant–microbe interactions to improve the sustainability and productivity of natural and agricultural systems [162,163]. The interrelatedness between biological and technical advancements has always had important implications on major breakthroughs and scientific advancements. We anticipate that the incredible complexity of plant holobionts will serve as fertile ground for new innovations in omics techniques and related technologies that will pioneer new advances in plant biology. Therefore, we hope that this perspective serves to stimulate new multidisciplinary research conducted in an environment that embraces the complexity of plant holobionts in order to catalyze new advancements to open up new biological questions for the plant–microbe research community.

This manuscript has been authored by UT-Battelle, LLC under Contract No. DE-AC05-00OR22725 with the U.S. Department of Energy (DOE). The United States Government retains and the publisher, by accepting the article for publication, acknowledges that the United States Government retains a non-exclusive, paid-up, irrevocable, worldwide license to publish or reproduce the published form of this manuscript, or allow others to do so, for United States Government purposes. The DOE will provide public access to these results of federally sponsored research in accordance with the DOE Public Access Plan.

## Figures and Tables

**Figure 1 microorganisms-10-02013-f001:**
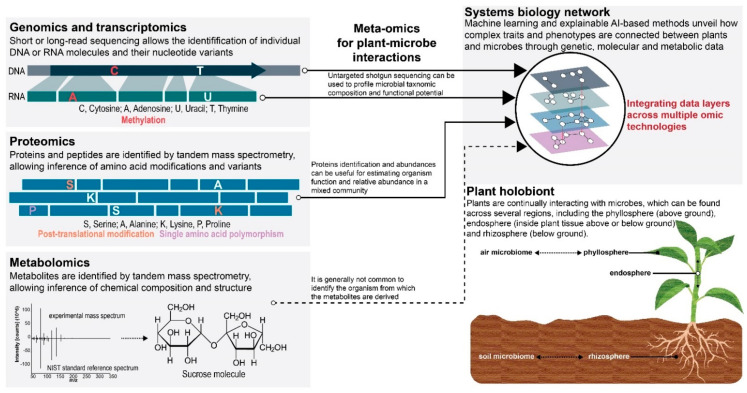
The application of major omic technologies and their integration provide complementary data necessary for dissecting complex traits and phenotypes associated with plant-microbe interactions. The promises and challenges of each omic technology will be influenced by the genetic, functional, and environmental complexity inherent to the biological system being studied.

**Figure 2 microorganisms-10-02013-f002:**
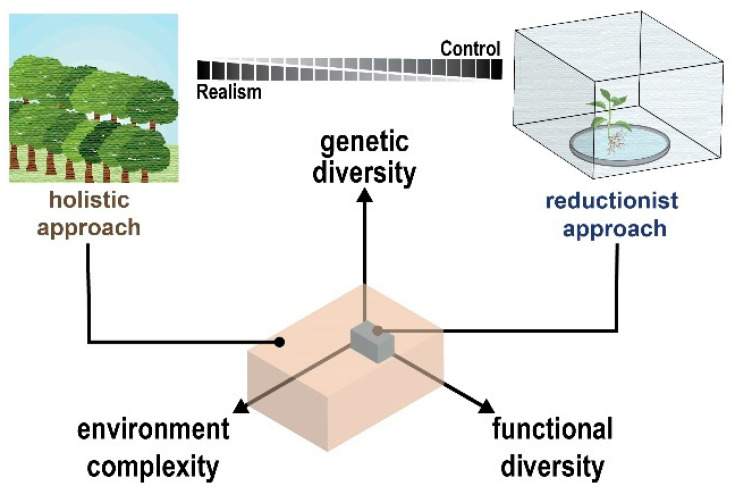
Dimensions of complexity and diversity in plant–microbe interactions. Studying plant–microbe interactions in both reduced and highly complex biological systems is necessary to obtain a complete understanding of these complex relationships. As complexity increases, the completeness and reliability of omic technologies will be challenged.

## Data Availability

Not applicable.

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
