# Peer review of "The Promises, Challenges, and Opportunities of Omics for Studying the Plant Holobiont"

_microorganisms, 2022, doi:10.3390/microorganisms10102013_

Round 1

Reviewer 1 Report (New Reviewer)

This review is focused on role of omics method to rich knowledge about plant microbe interaction. The authors clarify the role of machinery development in next generations and chromatography on genomics transcriptomic, proteomic and metabolomic data. Consequently improvement the understanding of plant microbe interaction. the manuscript is written in logic sequence and rich with information. really, i have no any concern and i would like thanks the authors for this useful review article

Author Response

We would like to thank the reviewer for their careful consideration and positive feedback.

Reviewer 2 Report (New Reviewer)

There is no mention of the likely role of viruses.  I can see, however, how this would add further complexity to what is already a very complex and dynamic set of interactions.

Author Response

We thank the reviewer for their thoughtful feedback and we agree that integrating viruses into this article would be challenging given the many technical and biological challenges that would need mentioning. We do look forward, though, to following this emerging area of plant-microbe research because relevant technologies are beginning to shed new lights on this topic. 

Reviewer 3 Report (New Reviewer)

The paper entitled "The promises, challenges, and opportunities of omics for studying the plant holobiont” present important information regarding what omics cab offer to unravell and to study the plant microbiome. The introduction is well-written and organized. The overall manuscrit is also well presented and supported by relevant references. I enjoyed reading it, as it is easly understandable and appealling for the readers.

Congratulations!

I have some minor corrections in the attached pdf, along with the following comment:

Line 46-50: This sentence is way too long. Please rewrite it.

Kind Regards

Author Response

We are thankful for the careful review and suggestions provided to improve the text. We have revised the text as suggested by the reviewer. 

This manuscript is a resubmission of an earlier submission. The following is a list of the peer review reports and author responses from that submission.

Round 1

Reviewer 1 Report

I have reviewed this manuscript carefully. After reading this manuscript, I realized that a similar but much better manuscript has recently been published in the same journal by Gamalero et al., 2022.  (Current Techniques to Study Beneficial Plant-Microbe Interactions). This review is much better than the one provided by authors. I could even see a similar figure in both reviews. 

All of the contents provided in the current review by authors are already mentioned in the previously published high quality review. Therefore, I could not gauarantee its publiocation. There is nothing new reported in the review and is just the mirror review of previously published one. I am sorry. 

Reviewer 2 Report

Dear Authors,

Thank you for the opportunity to review an excellent review “The promises, challenges, and opportunities of omics for 3 studying the plant holobiont.” It contributes to our understanding of each individual organism as a result of a very complex interaction between plant host and diverse microbes, occupying different ecological niches and parts of the plant. I really like the idea that there is basically no “pure” organism in the nature and in each case, we deal with the product of combined efforts between the organisms itself and its microbial “occupants”, which influence its growth and development and responses of the environment. Also, it is the first review for me, which considers all -omics.

Just a few remarks. Page 3, second paragraph. Since fungi are technically microbes, it would be worth mentioning early metagenomic surveys using not only 16S but also ITS and other fungal barcoding genes, I believe. Also, I see this manuscript as a more theoretical description of what such omics studies might reveal to us. It seems that your work would definitely benefit if you mentioned some major breakthroughs as the results of such studies. What did we actually learn from all these studies using various “-omics”? Just on the top of my head: genome reduction or gene family expansion as an adaptation to the environment, the race between fungal effectors and plant resistance genes in case of pathogens, conservation of the transcriptomic responses in the compatible versus incompatible host and ectomycorrhizal species pairings, perhaps.

Those remarks are not critical in any case. I really enjoyed reviewing your manuscript and the approach you took describing the new horizons in host-endosymbiont studies, which modern researchers have with NGS sequencing coupled with proteins and metabolites analysis. I wish to see your paper on the pages of the Microorganisms journal soon.

Sincerely,

Reviewer.

Reviewer 3 Report

I have read the manuscript carefully and found a similar article has recently been published in the same journal by Gamalero et al., 2022 (Current Techniques to Study Beneficial Plant-Microbe Interactions). All of the contents provided in the current review are already mentioned in the previously published review. Therefore, the review was not suitable for publication on  Microorganisms.